# Flexible and stable high-energy lithium-sulfur full batteries with only 100% oversized lithium

Jian Chang[1], Jian Shang[1], Yongming Sun[2], Luis K. Ono[3], Dongrui Wang[1], Zhijun Ma[1], Qiyao Huang[1], Dongdong Chen[1], Guoqiang Liu[1], Yi Cui[2,4], Yabing Qi [3] & Zijian Zheng [1]

Lightweight and flexible energy storage devices are urgently needed to persistently power wearable devices, and lithium-sulfur batteries are promising technologies due to their low mass densities and high theoretical capacities. Here we report a flexible and high-energy lithium-sulfur full battery device with only 100% oversized lithium, enabled by rationally designed copper-coated and nickel-coated carbon fabrics as excellent hosts for lithium and sulfur, respectively. These metallic carbon fabrics endow mechanical flexibility, reduce local current density of the electrodes, and, more importantly, significantly stabilize the electrode materials to reach remarkable Coulombic efficiency of >99.89% for a lithium anode and >99.82% for a sulfur cathode over 400 half-cell charge-discharge cycles. Consequently, the assembled lithium-sulfur full battery provides high areal capacity (3 mA h cm$^{-2}$), high cell energy density (288 W h kg$^{-1}$ and 360 W h L$^{-1}$), excellent cycling stability (260 cycles), and remarkable bending stability at a small radius of curvature (<1 mm).

[1] Laboratory for Advanced Interfacial Materials and Devices, Institute of Textiles and Clothing, The Hong Kong Polytechnic University, Hong Kong SAR, China. [2] Department of Materials Science and Engineering, Stanford University, Stanford, CA 94305, USA. [3] Energy Materials and Surface Sciences Unit, Okinawa Institute of Science and Technology Graduate University, 1919-1 Tancha, Onna-son, Okinawa 904-0495, Japan. [4] Stanford Institute for Materials and Energy Sciences, SLAC National Accelerator Laboratory, 2575 Sand Hill Road, Menlo Park, CA 94025, USA. Correspondence and requests for materials should be addressed to Z.Z. (email: tczzheng@polyu.edu.hk)

T he emergence of flexible and wearable electronics has a significant role in the realization of the Internet of Things, allowing the creation of intelligent fabric for body-worn or near-body sensors that can communicate with each other or with the internet[1–3]. To persistently power wearable devices, lightweight and flexible energy storage units with high energy density and electrochemical stability are in urgent need[4–7]. Rigid-typed lithium-ion batteries (LIBs) fabricated on metal foils are currently dominating battery technologies for portable electronics because of their relatively high energy density and long cycle-life. In the past decade, much effort has been devoted to developing flexible-typed LIBs; however, increasing the energy density while maintaining the light weight, flexibility and cycling stability of the devices remains challenging due to limited electrochemical capacity and high mass density of intercalation electrode materials. For example, the state-of-the-art flexible LIBs exhibit low energy density, i.e. <2 mW h cm$^{-2}$ and short cycling lifetime, i.e., <50 cycles[8–10].

Lithium-sulfur (Li-S) batteries show great promise as the next-generation high-energy-density batteries for flexible and wearable electronics because of their low mass densities (Li: 0.534 g cm$^{-3}$; S: 2.07 g cm$^{-3}$) and high theoretical capacities (Li: 3860 mA h g$^{-1}$; S: 1675 mA h g$^{-1}$)[11,12]. However, most reported Li-S batteries to date require the use of heavy Li foil anodes (~100 mA h cm$^{-2}$)[13–16]. The use of heavy Li foil as anode has led to several drawbacks, including (1) low Coulombic efficiency of Li anode (<90%) and the quick dry out of electrolyte[17,18], (2) low energy density due to the heavy weight of Li foil and the large amount of electrolyte[14–16], (3) easy formation of Li dendrites[19,20], and (4) poor mechanical flexibility[14,15,21].

To solve this problem, many research works focus on developing stable Li anode and S cathode with high Coulombic efficiency (CE). Several useful strategies have been reported to achieve stable Li metal anodes, including the design of artificial solid electrolyte interface (SEI) layers[18,22], the use of electrolyte additives[19,23,24], and the development of high-surface-area Li hosts[20,25–27]. On the cathode side, effective S hosts through physical confinement approaches[28–30] and/or chemical modification approaches using metal oxides[31], sulfides[32], and even nitrides[33] have been reported to reduce the shuttle effects of polysulfides. Regardless of this rapid progress in the past decade, the state-of-the-art still cannot avoid the use of a large excessive amount of Li, typically ~1500–15,000% oversize as compared with S cathode[31–35]. As a result, the reported Li-S batteries are still far from the viable commercial target to replace Li-ion technology[36–38].

It has been pointed out lately that a critical challenge for Li-S batteries is achieving high energy density and high stability in a full cell with the use of a Li anode that is only 6 mA h cm$^{-2}$ in size (~100% oversize)[38]. This is also of remarkable importance from a sustainability point of view, as the production of Li is environmentally unfriendly and expensive[39]. Until now, high energy density and stable cycling has not been achieved in a flexible Li-S full cell with a limited source of Li.

To address this critical challenge, we report here a highly flexible, stable and high energy density Li-S full battery, with the use of only 100% excess of Li. The key is to deposit Li or S onto rationally designed metal-coated carbon fabrics (CFs). Specifically, Li deposited on Cu-coated CF (Li/CuCF) is used as an anode, while a graphene/sulfur mixture on Ni-coated CF (NSHG/S$_8$/NiCF) is used as a cathode. The fabric structure simultaneously endows mechanical flexibility and reduces local current density of the electrodes. More importantly, the metal coating on CF significantly stabilizes the electrode materials to reach remarkable CE. On the anode side, the Cu coating renders uniform deposition of Li nanosheets instead of dendrites and leads to an average

CE > 99.89% over 400 charge-discharge cycles. On the cathode side, the Ni coating can catalytically accelerate polysulfides reduction and strongly anchor Li$_2$S, which leads to an excellent capacity retention > 99.82% over 400 cycles. Ultimately, mechanically robust Li-S full cells with high energy density (6.3 mW h cm$^{-2}$), high areal capacity (3 mA h cm$^{-2}$), large current density (2 mA cm$^{-2}$) and excellent cycling stability (capacity retention per cycle: 99.89% for 260 cycles) have been successfully achieved. The Li-S full cells could maintain stable charge/discharge characteristics over 200 cycles and power large LED screens to display for tens of minutes even when repeatedly bent at small radii of curvature (<1 mm).

## Results

**Design and fabrication of lithium-sulfur full batteries**. Fabric-based batteries can effectively accommodate the complex deformations induced by body motion and are easy to integrate into the ordinary cloth. To construct Li-S fabric batteries, we firstly fabricate metal-coated CF for use as current collectors. CFs are known to be lightweight and flexible, but their high electrical resistance[40], chemical instability[41–43], and poor affinity for lithium and sulfur[27–29,33] are far from being the ideal host. Therefore, a uniform layer of chemically stable and highly conductive Cu (for anode) or Ni (for cathode) is deposited onto CFs via a modified polymer-assisted metal deposition (PAMD) method[44–46]. In brief, CFs were activated through an acid treatment to render the surface hydrophilic, followed by surface-initiated polymerization to form a 10-nm-thick poly[2-(methacryloyloxy) ethyl] trimethyl ammonium chloride (PMETAC) coating (Fig. 1a). The PMETAC layer acted as an interfacial layer to immobilize Pd moieties through the quaternary ammonium groups, after which an electroless deposition (ELD) of dense metal nanoparticles onto the fabrics was carried out from the PMETAC layer (Fig. 1b, c, and Supplementary Fig. 1 and Supplementary Fig. 1). Both Ni-deposited carbon fabric (NiCF) and Cu-deposited carbon fabric (CuCF) show exceedingly low sheet resistance of 0.48 and 0.1 Ω cm$^{-2}$, respectively, which are one order of magnitude lower than that of pristine CF (3.04 Ω cm$^{-2}$, Fig. 1d), and the resistance remains unchanged over 5000 bending cycles. Furthermore, these metallic fabrics exhibit much higher tensile and compression strength than that of CFs without obvious density increase (Fig. 1d, Supplementary Fig. 3a, and 3b), which is in accordance with the wear-resistant and lightweight requirements for wearable devices. The remarkable properties can be ascribed to the uniform polymer interfacial layer (i.e., PMETAC) that bonds the metal to the CF surface.

Subsequent to the metal deposition, a certain amount of Li metal was electrochemically plated on CuCF to yield the Li/CuCF anode. On the other hand, a slurry mixture containing nitrogen and sulfur heavily doped graphene (NSHG), Super P (a type of carbon black), and S$_8$ was coated on NiCF to yield the NSHG/S$_8$/NiCF cathode (Fig. 1a). Finally, the two pieces of fabric electrodes, together with a commercial separator membrane and ether-based electrolyte, were assembled into a soft full cell and sealed in an aluminum encapsulation.

**Plating and stripping behavior of lithium anode**. As shown in the cross-sectional scanning electron microscopy (SEM) images at different positions of the anode, the electrodeposition of Li metal was very uniform (Supplementary Fig. 4). The Li metal preferentially deposits on the top part of CuCF due to the shorter Li$^+$ diffusion pathway, as opposed to the bottom part. With increasing the areal capacity of Li/CuCF anode, the deposition of Li metal gradually expands from the top to the bottom of CuCF (Fig. 2a and Supplementary Fig. 5), reaching a thickness of ~80

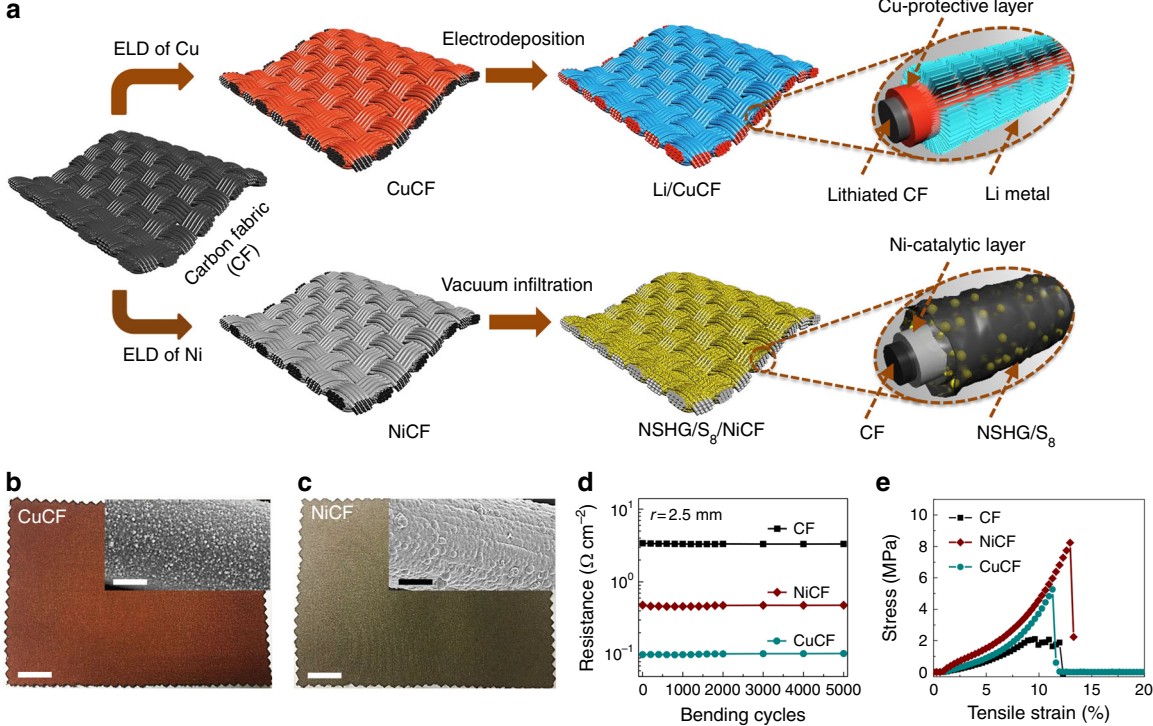

**Fig. 1** Schematic of the fabrication process and design principle for the electrodes. **a** To simultaneously endow mechanical durability and cycling stability of Li-S batteries, chemically stable and highly conductive Cu or Ni thin layers are uniformly coated onto carbon fabrics (CuCF or NiCF) via a scalable polymer-assisted metal deposition (PAMD) method. After electrodeposition, the achieved Li/CuCF anode with Cu-protective layers and lithiated carbon could stabilize Li metal deposition and offset the irreversible Li loss during plating/stripping process. After vacuum infiltration, the NSHG/$S_8$/NiCF cathode (NSHG is nitrogen and sulfur heavily doped graphene) with Ni-catalytic layers could simultaneously speed up the redox kinetics of soluble polysulfides and solid $Li_2S$. **b**, **c** Digital images of scalable CuCF and NiCF (scale bar = 3 cm). The corresponding enlarged metallic fibers are also observed using scanning electron microscopy (SEM) images, as shown in the inset (scale bar = 3 μm). **d** Sheet resistances of various fabrics (CF, CuCF, and NiCF) as a function of bending cycles at a given bending radius ($r$ = 2.5 mm). **e** Typical tensile stress-strain curves of CF, CuCF, and NiCF. (ELD is electroless deposition)

μm when the amount of deposition was 10 mA h cm$^{-2}$. For this thick sample, Li metal is plated onto the surface of metallic fibers and the space among them at the upper position of the fabric (P1, Fig. 2a). At the middle position of the fabric (P2), each metallic fiber is coated by Li metal and some gaps are filled. At the lower position (P3), no Li metal is observed.

The CuCF possesses an ultrahigh ability to stabilize Li metal during its electroplating and stripping process. As a proof-of-concept, we firstly analyzed the nucleation behavior of Li metal during the 1st electroplating process, in which the nucleation overpotential of Li is defined as the difference between the sharp tip voltage and the later stable mass-transfer overpotential. When plating Li onto bare Cu foil, a large nucleation overpotential of 170 mV is observed (Fig. 2b), which indicates the unfavorable interaction between Cu foil and Li metal. In contrast, the nucleation overpotential of Li metal on CuCF is only 15 mV. The electrodeposition on CuCF exhibits a two-step process: an initial Li$^+$ intercalation into CuCF from 0.75 V (vs. Li/Li$^+$) and a subsequent Li metal deposition on the Cu surface at nearly 0 V (Supplementary Fig. 6). The initial Li$^+$ intercalation step is believed to form a LiC$_x$ complex with the carbon fibers, which turns CuCF into a highly lithiophilic substrate for the subsequent Li deposition. In addition, the Cu nanoparticles, which fully cover the surface of CuCF, may also serve as nucleation centers to further enhance the affinity to Li metal. Uniform spherical particles and partial ultrathin nanosheets of metallic Li are observed inside-out the CuCF after the electroplating (Fig. 2c, d).

Apart from the ultralow nucleation overpotential at the 1st electroplating process, Li/CuCF also maintain a significantly low mass-transfer overpotential in a long-term repeating stripping/

plating of Li metal. Here, coin cells made of one pair of Li/CuCF anodes (electrode capacity: 6 mA h cm$^{-2}$) are stripped and plated in a partial capacity of 2 mA h cm$^{-2}$ at a current density of 1 mA cm$^{-2}$. It is found that the Cu nanoparticles on the surface of CuCF can also confine the deposits of metallic Li and allow the continuous formation of Li nanoflakes on CuCF during the stripping/plating process (Fig. 2e, f). Namely, the lithiated CuCF with high-porosity can serve as a stable cage for accommodating Li metal inside fabric structure, further stabilizing the SEI layer. The spherical particles of Li metal formed at the 1st electroplating process gradually transform into uniform nanoflakes. The Li nanoflakes significantly reduce the mass-transfer overpotential of Li metal by allowing much faster Li$^+$ transport and Li metal deposition, which lead to steadily low overpotential. As shown in Fig. 2g and h, the overpotential of Li/CuCF symmetric cell starts at a very low value of ~30 mV. It continues to decrease until reaching ~20 mV at the 10th cycle and keeps constant for 120 cycles. In comparison, when one pair of pure Li foils or Li on Cu foils (Li/Cu foil) are assembled into symmetric coin cells, obvious dendrite formation was observed (Supplementary Fig. 8). The fragment and delamination of Li foil after cycling adversely increase the overpotential and simultaneously speed up the Li loss. For Li/Cu foil, the uncontrolled mossy deposits of Li metal lead to the violent fluctuation of overpotential at the 15th cycle and the sudden jump at the 35th cycle.

The ability to stabilize Li metal with CuCF becomes even more prominent when the striping/plating process is carried out at high rate. For example, when the current density is increased to a very high rate of 5 mA cm$^{-2}$, Li/CuCF still shows a successively reduced overpotential at the early stage and keeps a very low

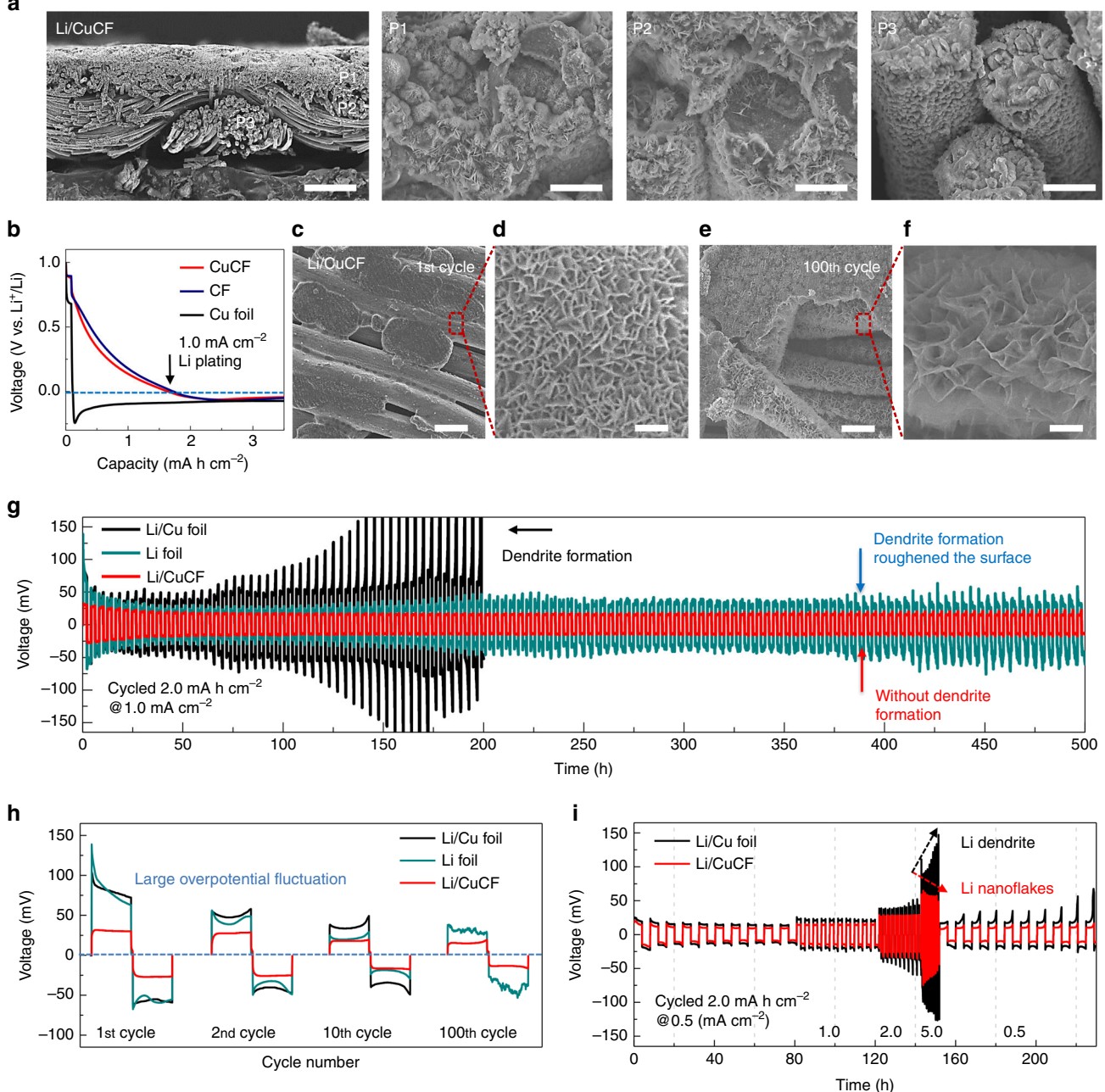

**Fig. 2** Plating and striping behavior of lithium anode. **a** Low and high magnification scanning electron microscopy (SEM) images of lithiated copper-deposited carbon fabric (Li/CuCF) anode in the through-thickness direction (scale bar for **a** is 100 and 5 μm, respectively). The upper position (P1), middle position (P2) and lower position (P3) of the fabric are shown. **b** Nucleation behavior of Li metal onto various hosts (copper-deposited carbon fabric (CuCF), carbon fabric (CF), and Cu foil) at 1 mA cm$^{-2}$ during the 1st electroplating process. **c–f** Low and high magnification SEM images of Li/CuCF anode after the 1st and 100th plating/striping cycle at 1 mA cm$^{-2}$ (scale bar for **c** and **e** is 100 μm; scale bar for **d** and **f** is 0.2 and 2 μm, respectively). **g** Galvanostatic plating/stripping profiles in Li/Cu foil, Li foil, and Li/CuCF symmetric cells at 1 mA cm$^{-2}$. **h** Galvanostatic plating/stripping profile of various Li anodes at 1st, 2nd, 10th, 100th cycle. **i** Rate performance comparison between Li/CuCF and Li/Cu foil at various current densities from 0.5 mA cm$^{-2}$ to 5 mA cm$^{-2}$. The areal capacity of Li metal anodes is 10 mA h cm$^{-2}$ (**a**) and 6 mA h cm$^{-2}$ (**b–i**), respectively

value of ~50 mV (Fig. 2i). The Li/Cu foil, however, results in drastically increased overpotential over 150 mV just after 10 cycles.

**Coulombic efficiency and cycling stability of lithium anode.** For Li metal anode, the primary challenge is to reduce the Li loss during the cycling, i.e., to achieve a high CE to ensure a long cycle life. The cycled Li loss primarily originates from two unstable

interfaces: (i) SEI layers which repeatedly break and repair due to the dendrite formation, and (ii) the interfacial side reaction between Li and the substrate surface, which repeatedly occurs during cycling.

A high CE of the Li/CuCF anode can be expected due to the high stability of Li on Li/CuCF and a low operation overpotential discussed above. As proof-of-concept, we recorded the CEs of various Li anodes including Li/CuCF, Li/CF, and Li/Cu foil during the electrochemical cycling against Li foil, where 3.5 mA h

$cm^{-2}$ of Li is fully plated and stripped at a current density of 1 $mA\ cm^{-2}$ (Fig. 3a). For Li/Cu foil, the CE rapidly drops to below 90% after only 6 cycles, resulting from the dendrite formation and SEI damage. Compared to Li/Cu foil, Li/CF exhibits a higher average CE of 98.2% during the first 40 cycles, indicating that the three-dimensional fabric structure with high surface area can reduce the effective current density and stabilize the SEI layer. Unfortunately, Li/CF suffers from a cycling deterioration after the 40th cycle, which is attributed to the Li-ether co-intercalation and subsequent structural damage[26,42,47]. Li/CuCF shows a much higher average CE of 99.54% and much longer cycling stability than that of CF and Cu foil. This can be ascribed to the fact that the Cu coating can effectively prevent side reaction between Li and CF surface.

To more accurately evaluate the CE and deeply investigate the storage mechanism of Li metal anodes, a $Li_4Ti_5O_{12}$ (LTO) counter electrode (1 $mA\ h\ cm^{-2}$) is paired with an oversized Li/CuCF anode (3.5 $mA\ h\ cm^{-2}$) (Fig.3b). The total capacity of Li/CuCF anode includes the plated Li metal capacity of 2 $mA\ h\ cm^{-2}$ and the intercalated $Li^+$ capacity of 1.5 $mA\ h\ cm^{-2}$. The use of the LTO counter electrode ensures that all the cycled Li loss originates from Li/CuCF anode because LTO has no prestored Li. It is observed that there is a large Li loss of 0.18 $mA\ h\ cm^{-2}$ due to the formation of SEI in the 1st cycle (Fig. 3b). From the 2nd cycle, the capacity of the cell decades slowly until the 140th cycle. When the capacity of Li metal is in excess to that of LTO, the striping process only occurs to the plated Li metal (Fig. 3c, d and Supplementary Fig. 9). This process is defined as Stage I. The average CE of Li metal in Stage I is calculated to be 99.42% in Li/CuCF anode. In Stage II starting from the 141st cycle, the striping process occurs to both the Li metal and the intercalated $Li^+$ due to

the less capacity of Li metal than that of LTO. In Stage III starting from the 300th cycle, the capacity of the cell becomes very stable and no obvious drop is observed. The CE of Stage III is 99.99%, which is attributed to the highly reversible $Li^+$ insertion/desertion of the CuCF. Therefore, the capacity depletion in Stage II can be ascribed to the loss of Li metal, and the average CE of Li metal is determined to be 99.89% in Li/CuCF anode. Following the same testing conditions, the reference cell made of Li/CF//LTO shows a much shorter Stage I until only 50th cycle (Fig. 3d), which corresponds to a low CE of 98.36%. The reference cell made of Li/Cu foil//LTO with similar capacity loading decades even much more rapidly and loses all the capacity at the 70th cycle, which corresponds to a very poor CE of 71.7%. Again, the results indicate that the three-dimensional metallic CF current collector can not only reduce the effective current density and stabilize the SEI layer, but also prevent the side reaction between Li and CF surface. Furthermore, the partially intercalated $Li^+$ in Li/CuCF anode may offset the irreversible Li loss of cycled Li metal and further extend the cycling performance of Li/CuCF.

**Electrochemical properties of sulfur cathode.** The as-prepared NiCF was immersed into a pre-mixed slurry containing 70 wt% $S_8$, 17.5 wt% Super P, and 12.5 wt% NSHG and then dried in a vacuum oven. The NSHG is synthesized through a modified procedure[48] and the detail characterization is listed in Supplementary Fig. 10. From the SEM and transmission electron microscopy (TEM) characterizations (Fig. 4a, d), the fiber surfaces are uniformly and densely wrapped with the well-mixed NSHG/Super P/$S_8$ nanocomposites after the coating process. From the electron probe micro-analyzer (EPMA) and

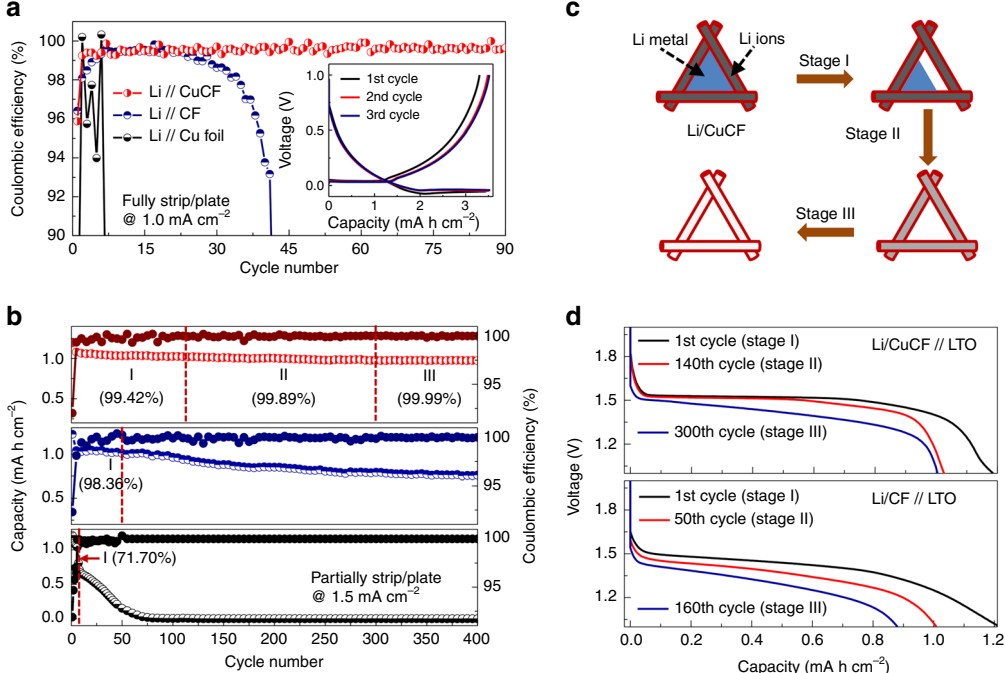

**Fig. 3** Cycling stability and storage mechanism of lithium anode. **a** The Coulombic efficiency (CE) of various Li anodes including lithiated copper-deposited carbon fabric (Li/CuCF), lithiated carbon fabric (Li/CF) and lithiated copper foil (Li/Cu foil) during cycling against Li foil, where 3.5 $mA\ h\ cm^{-2}$ of Li is fully plated and stripped at 1 $mA\ cm^{-2}$. The profiles of areal capacity versus voltage for Li/CuCF anode are recorded in the inset. **b** Areal capacities and corresponding CE values of Li/CuCF, Li/CF, and Li/Cu foil anode- $Li_4Ti_5O_{12}$ (LTO) cells for 400 cycles are plotted at a current density of 1.5 $mA\ cm^{-2}$ from top to bottom, respectively. The cycling process is involved in three stages: Stage I represents the cycling process that only occurs to the plated Li metal; Stage II represents the cycling process that occurs to both the plated Li metal and the inserted $Li^+$; Stage III represents the cycling process that only occurs to the inserted $Li^+$. **c** Scheme of three cycling Stages (I, II, and III) for Li/CuCF anode. **d** Galvanostatic discharge profiles of Li/CuCF-LTO and Li/CF-LTO cells at 1.5 $mA\ cm^{-2}$ for various cycling stages. The areal capacity of Li metal anodes is 3.5 $mA\ h\ cm^{-2}$ in all cases

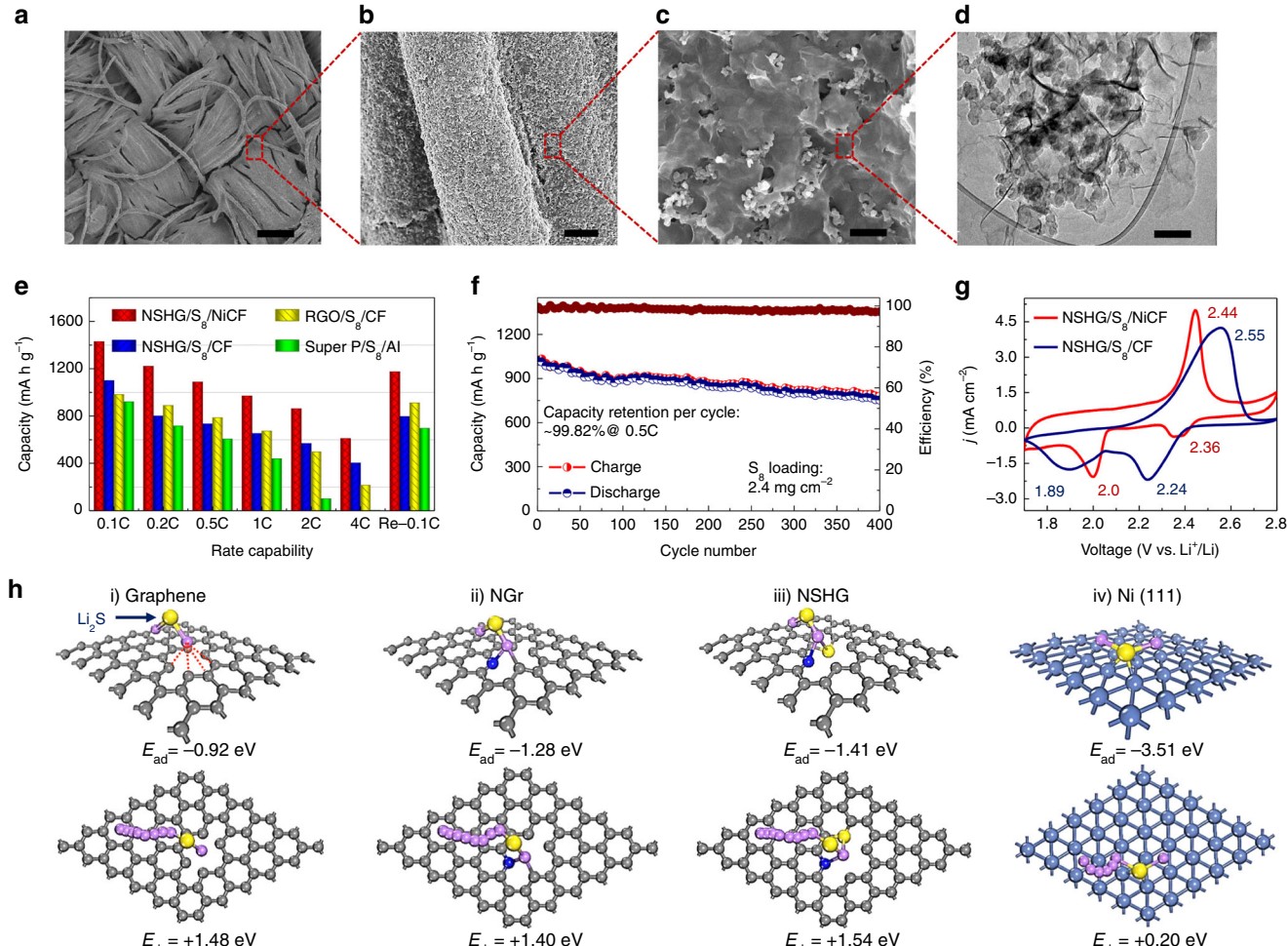

**Fig. 4** Cycling stability and rate capability of sulfur cathode. **a–c** Low and high magnification scanning electron microscopy (SEM) images of the NSHG/$S_8$/Ni cathode (NSHG is nitrogen and sulfur heavily doped graphene). The sulfur cathode was obtained by direct infiltration of NSHG/$S_8$ hybrid inks into the nickel-deposited carbon fabric (NiCF), followed by drying at 60 °C in a vacuum chamber. Scale bar for **a–c** is 100 μm, 5 μm, and 500 nm, respectively. **d** Transmission electron microscopy (TEM) images of NSHG/$S_8$ cladding layers, extracted from NSHG/$S_8$/Ni cathode (scale bar = 200 nm). **e** Specific capacities at various rates for sulfur cathodes, including NSHG/$S_8$/NiCF, NSHG/$S_8$/CF (CF is carbon fabric), RGO/$S_8$/NiCF (RGO is reduced graphene oxide) and Super P/$S_8$/Al electrodes (Super P is a type of carbon black). **f** Specific capacities, and corresponding CE values of the optimized NSHG/$S_8$/NiCF cathode were obtained at 2 mA cm$^{-2}$ for 400 cycles. **g** Cyclic voltammetry curves of NSHG/$S_8$/NiCF and NSHG/$S_8$/CF at a scan rate of 0.2 mV s$^{-1}$ in a potential window from 1.7 to 2.8 V. **h** Theoretical calculation of the adsorption and decomposition energies of $Li_2S$ on the surface of RGO, nitrogen-doped graphene (NGr), NSHG, and the thin slab of Ni (111). Notably, the thin slab of Ni (111) with a large adsorption energy of −3.51 eV and a tiny decomposition energy barrier of 0.2 eV for $Li_2S$ facilitates the rapid oxidation of $Li_2S$ back to sulfur (carbon is gray, sulfur is yellow, lithium is purple, nitrogen is blue, nickel is light blue)

thermogravimetric analysis (TGA) (Supplementary Fig. 11a, b), the precise content of $S_8$ is determined to be 75 wt% in the nanocomposite coating. The highly polar NSHG serves as a conductive binder to strongly immobilize sulfur onto the surface of each metallized fiber. The Super P addition mainly contributes to the prevention of NSHG restacking. Finally, the NSHG/$S_8$/NiCF cathode with various sulfur loadings can be prepared by tuning the amount of NMP solvent due to the varying viscosity of the ink (Supplementary Fig. 11c–e). Notably, the porous NiCF can uniformly adsorb the sulfur mixture over a large lateral area and the cross-section direction due to the capillary force (Supplementary Figs. 11–14). At a relatively low mass loading of 1.4 mg cm$^{-2}$, the fiber surfaces are coated with sulfur slurry. At 3.2 mg cm$^{-2}$, the space between the fibers is partially filled. As mass loading increases to 5.6 mg cm$^{-2}$, most spaces are filled up.

The NSHG/$S_8$/NiCF cathode (vs. Li foil) exhibits high capacities at a wide range of discharging rates ranging from 0.1C to 4C (1C = 2.3 mA cm$^{-2}$). For example, the initial capacity

at 0.1C, 0.5C, and 4C reach 1427, 1016, and 614 mA h g$^{-1}$, respectively (Fig. 4e). The capacities at all rates are much superior to reference samples made with NSHG/$S_8$/CF, RGO/$S_8$/CF (RGO = reduced graphene oxide) and super P/$S_8$/Al cathodes (Fig. 4e, Supplementary Figs. 15 and 16). More importantly, the capacity retention per cycle of NSHG/$S_8$/NiCF reaches 99.82% over 400 cycles with a high mass loading of 2.4 mg cm$^{-2}$ (Fig. 4f). A stable areal capacity of 3.51 mA h cm$^{-2}$ is obtained at a sulfur mass loading of 4.2 mg cm$^{-2}$ (Supplementary Fig. 15i). After cycling, no obvious morphological change of the NSHG/$S_8$/NiCF cathodes was observed from topographical and cross-section SEM analysis (Supplementary Figs. 17 and 18).

The excellent rate capability, cycling stability, and high areal capacities of NSHG/$S_8$/NiCF cathode are ascribed to three main factors as follows. (1) The incorporation of high-surface-area CF could improve the accessibility of electrolytes to the electrodes, e.g., RGO/$S_8$/CF electrodes exhibit much better capacities compared to Super P/$S_8$/Al at all rates (Fig. 4e). (2) The addition

of NSHG not only serves as excellent binders to immobilize solid sulfur onto the metallic fabric, but also takes advantage of the hetero-atoms (N and S) doping sites to chemically absorb soluble polysulfides. That is, NSHG/$S_8$/CF electrodes further enhance the capacities at high rates compared to RGO/$S_8$/CF. (3) Most importantly, the Ni coating supported with high-surface-area CF effectively catalyze the reduction/oxidation of soluble polysulfides and the absorption/decomposition of the $Li_2S$ end-product to improve the utilization of sulfur/$Li_2S$. The rapid growth of uniform sulfur/$Li_2S$ nanoparticles within conducting NSHG matrix onto NiCF illustrates the catalytical role of Ni-metallic layers.

To further prove the catalytic property of Ni surface, we tested the electrochemical activity of the electrodes by cyclic voltammetry (CV) at a potential window of 1.7–2.8 V and a scan rate of $0.2 \, mV \, s^{-1}$. Both NSHG/$S_8$/NiCF and NSHG/$S_8$/CF electrodes exhibit two cathodic peaks and one anodic peak (Fig. 4g). The two representative cathodic peaks originate from the reduction of elemental sulfur to soluble lithium polysulfides ($Li_2S_x$, $4 \leq x \leq 8$) at the higher potential and the formation of insoluble lithium sulfides ($Li_2S_2/Li_2S$) at the lower potential, respectively. When scanning back, one anodic peak shows the oxidation of $Li_2S$ to $S_8$. The lower oxidation potential and higher reduction potential always reflect lower polarization and faster redox rates in electrochemical systems. Compared to NSHG/$S_8$/CF electrodes, NSHG/$S_8$/NiCF electrodes reveal the large upshift (2.24 V → 2.36 V; 1.89 V → 2.00 V) of the two reduction peaks and the downshift (2.55V → 2.44V) of the oxidation peak, which indicates that the introduction of polar Ni coating on CF can efficiently catalyze the reduction/oxidation of soluble polysulfides. In addition, NSHG/$S_8$/NiCF electrodes also exhibit much narrower and sharper redox peaks than that of NSHG/$S_8$/CF and Super P/$S_8$/Al, reconfirming the reduced polarization and rapid redox kinetics (Supplementary Fig. 19).

Stable and high capacity sulfur cathodes require not only rapid reduction/oxidation kinetics of soluble polysulfides, but also efficient absorption/decomposition of solid $Li_2S$[49]. In principle, highly reversible charge/discharge characteristics of the sulfur/$Li_2S$ cathode requires the large adsorption energy and small decomposition energy of $Li_2S$. From the density functional theory (DFT) simulation, we also found that of the Ni surface can immobilize lithium sulfides ($Li_2S$) more effectively in addition to the catalytic effect of polysulfides. It is known that carbon surface has low affinity to $Li_2S$. Therefore, the absorption and decomposition energies of on various modeling slabs including RGO, nitrogen-doped graphene (NGr), NSHG, and Ni (111) were well-studied (Fig. 4h and Supplementary Table 2). Compared to that of RGO (0.92 eV) and NGr (1.28 eV), the higher binding affinity of NSHG (1.41 eV) with $Li_2S$ indicates the incorporation of N and S heteroatoms can improve the deposition of polar $Li_2S$ onto the nonpolar graphene surface. Compared to RGO, NGr, and NSHG, the thin slab of Ni (111) exhibits the much stronger binding energy (3.51 eV) with polar $Li_2S$ and subsequently forms three S-Ni chemical bonds. On the other hand, the decomposition energies of $Li_2S$ on the surface of RGO, NGr, NSHG, and the thin slab of Ni (111) were calculated to be 1.48, 1.40, 1.54, and 0.20 eV, respectively. Notably, an intact $Li_2S$ molecule is decomposed into LiS and $Li^+$ ($Li_2S \rightarrow LiS + Li^+ + e^-$). As shown in Fig. 4h, the $Li^+$ rapidly diffuses far from the sulfur atom of $Li_2S$ and finally stabilizes at the center of C/Ni rings, which is accompanied by the breakage of the Li-S bond. Compared to graphene materials, the thin slab of Ni (111) exhibits the lowest decomposition energy barrier, suggesting efficient catalytic effects of Ni on $Li_2S$ decomposition. In other words, the efficient absorption and decomposition capability of the Ni layer significantly facilitates the oxidation of $Li_2S$ back to sulfur. Therefore, the Ni-coated

layer of NSHG/$S_8$/NiCF can simultaneously speed up the redox kinetics of soluble polysulfides and solid $Li_2S$, subsequently resulting in high sulfur recycling utilization.

**Cycling stability and flexibility of lithium-sulfur full batteries.** To examine the structural stability of both electrodes during mechanical deformations, sheet resistance of anodic Li/CuCF and cathodic NSHG/$S_8$/NiCF with various mass loadings have been provided as a function of bending cycles at a given bending radius $r = 2.5$ mm (Fig. 5a, b). The sheet resistance of both electrodes increases very slightly with the increase in mass loading of the electrode. The sheet resistances of both electrodes do not show obvious increase over 2000 bending cycles, which is well contrasted with Li foil. No delamination of the electrode materials from the metallized fabrics are observed (Supplementary Fig. 20 and Fig. 21). Therefore, the high flexibility, high capacity and electrochemical stability of Li/CuCF and NSHG/$S_8$/NiCF cathode are highly suitable for making high energy flexible Li-S full cells. As proof-of-concept, we stacked the two fabric electrodes with a polypropylene separator to fabricate the full battery (Fig. 5c). NSHG/$S_8$/NiCF cathodes with different sulfur loadings of 1.4 and $3.2 \, mg \, cm^{-2}$ are paired with a limited amount of Li/CuCF anode ($6 \, mA \, h \, cm^{-2}$) (Fig. 5d). The electrochemical properties of the as-made Li-S full batteries under different current densities are summarized in Table 1. It should be noted that the gravimetric and volumetric densities are calculated based on the total weight and volume of the entire battery including current collector, electrode, and separator.

Importantly, both batteries with high and low mass loading show remarkable cell capacity and stability. For example, the Li-S battery with $3.2 \, mg \, cm^{-2}$ sulfur cathode exhibits high capacities of 3.8, 3, and $2.4 \, mA \, h \, cm^{-2}$ at 0.5, 1, and at $2 \, mA \, cm^{-2}$, respectively. At a practical current density of $1 \, mA \, cm^{-2}$, the cell provides high areal energy density of $6.3 \, mW \, h \, cm^{-2}$, gravimetric energy density of $288 \, W \, h \, kg^{-1}$, and volumetric energy density of $360 \, W \, h \, L^{-1}$ (Table 1). To the best of our knowledge, all of the corresponding battery metrics are much higher than the state-of-the-art flexible LIBs and recently reported Li foil-based flexible Li-S batteries (Supplementary Table 3 and Table 4). Most importantly, the Li-S full battery with limited Li anode could be cycled up to 260 cycles with a capacity retention of 99.89% per cycle at a high current density of $2 \, mA \, cm^{-2}$, which is even comparable to inflexible Li-S batteries with a high sulfur loading[50–52]. The Li-S battery with sulfur loading of $1.4 \, mg \, cm^{-2}$ also provides a good capacity of $1 \, mA \, h \, cm^{-2}$ and an even higher capacity retention of 99.92% per cycle over 260 cycles (Supplementary Fig. 22).

The Li-S full battery is ideal for flexible and wearable applications. To showcase the capability, two batteries of a size of $4 \, cm^{-2}$ are connected in series to yield an open circuit voltage of 4.2 V and a high areal capacity of $4 \, mA \, h \, cm^{-2}$. The tandem cell is used to power a screen consisting of 264 light-emitting diodes (LEDs) (trigger voltage: 3.7 V; size: $10 \times 3 \, cm^{-2}$), which display a clear caption of "Li S Fabric" (Fig. 5e and Supplementary Fig. 23). The LED screen keeps lighting with stable brightness during repeated bending at small radii of curvatures of 7.5 and 5 mm for tens of minutes. The charge/discharge characteristics of the battery is recorded at $1 \, mA \, cm^{-2}$ for 150 cycles, during which 200 repeating bends at 5 mm radius are carried out. Again, the high capacity retention of ~99.5% and small fluctuations during mechanical bending indicate the excellent mechanical stability of Li-S full batteries. To further examine the mechanical robustness of our textile-based Li-S full cells, more severe folding deformation ($r < 1$ mm) at higher current density ($2 \, mA \, cm^{-2}$) was conducted. As shown in Fig. 5f, 40 folds of the Li-S full batteries were carried out during the 50

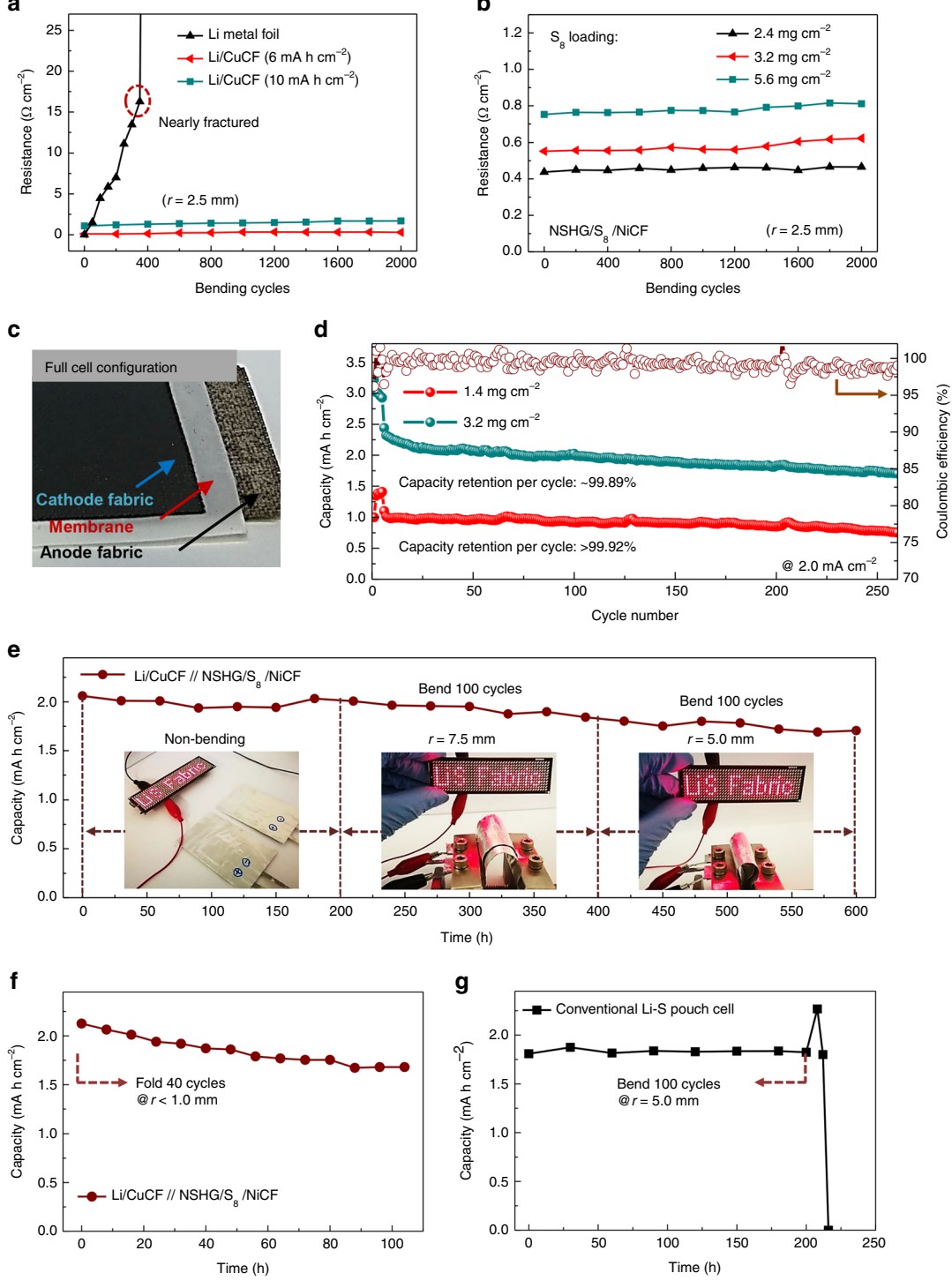

**Fig. 5** Cycling stability and mechanical durability of lithium-sulfur full batteries. **a**, **b** Sheet resistances of the electrodes (lithiated copper-deposited carbon fabric, Li/CuCF and NSHG/S$_8$/NiCF where NSHG is nitrogen and sulfur heavily doped graphene and NiCF is nickel-deposited carbon fabric) as a function of bending cycles at a given bending radius ($r = 2.5$ mm). **c** Digital picture of the inner cell configuration in full-cell Li-S batteries. A flexible microporous separator is used for separating two kinds of composite fabrics (NSHG/S$_8$/NiCF as cathode and Li/CuCF as anode). **d** Cycling stability of Li-S full batteries at high current densities of 1 mA cm$^{-2}$ and 2 mA cm$^{-2}$. The sulfur cathode of NSHG/S$_8$/NiCF with different sulfur loadings of 1.4 mg cm$^{-2}$ and 3.2 mg cm$^{-2}$ was paired with a limited amount of Li/CuCF (6 mA h cm$^{-2}$) to study the cycling performance. **e** Areal capacities of our Li-S full cells with a size of 4 cm$^{-2}$ are recorded at 1 mA cm$^{-2}$ for 150 cycles, during which 200 repeating bents at 5 mm radius are carried out. To demonstrate flexible and wearable applications, two batteries of a size of 4 cm$^{-2}$ are connected in series to yield an open circuit voltage of 4.2 V and a high areal capacity of 4 mA h cm$^{-2}$. The tandem cell can be used for powering 264 LEDs-array screens for tens of minutes. The array screen can display a clear caption of "Li S Fabric", although the Li-S batteries as a power supplier were harshly bended at various bending radius of (i) $r = \infty$, (ii) $r = 7.5$ mm, (iii) $r = 5$ mm, as shown in the inset. **f** Areal capacities of Li-S full cells with a size of 4 cm$^{-2}$ are recorded at 2 mA cm$^{-2}$ for 50 cycles, during which 40 repeating folds are performed at a folding radius of <1 mm. **g** Areal capacities of conventional Li foil-based Li-S pouch cells with a size of 4 cm$^{-2}$ are recorded at 1 mA cm$^{-2}$ for 53 cycles and 100 repeating bends at 5 mm radius are conducted after the 50th cycle

**Table 1 Performance metrics of our flexible Li-S full batteries**

| Current (mA cm$^{-2}$) | Cell weight (mg cm$^{-2}$) | Cell volume (cm$^3$) | Areal capacity (mA h cm$^{-2}$) | Areal energy (mW h cm$^{-2}$) | Gravimetric energy (W h kg$^{-1}$) | Volumetric energy (W h L$^{-1}$) |
|---|---|---|---|---|---|---|
| 0.5 | 19.28 | 0.0175 | 1.4 | 2.94 | 152 | 168 |
| 1.0 | 19.28 | 0.0175 | 1.0 | 2.1 | 109 | 120 |
| 0.5 | 23.28 | 0.0175 | 3.8 | 8.0 | 344 | 457 |
| 1.0 | 21.88 | 0.0175 | 3.0 | 6.3 | 288 | 360 |
| 2.0 | 21.88 | 0.0175 | 2.4 | 5.04 | 230 | 288 |

Experimentally, the electrode mass of lithiated copper-deposited carbon fabric (Li/CuCF) anode with high areal capacities of 6 mA h cm$^{-2}$ was measured to be 8.6 mg cm$^{-2}$. To increase the areal capacity of Li-S full batteries, the NSHG/S$_8$/NiCF cathode (NSHG is nitrogen and sulfur heavily doped graphene and NiCF is nickel-deposited carbon fabric) with various electrode mass of 10.5, 13.1, and 14.5 mg cm$^{-2}$ was prepared to pair with the Li/CuCF anode. The measured thickness of metallic fabrics/electrodes is ~150 μm under standard stress (400 N cm$^{-2}$, a pressure used for the compression of standard coin cells). The thickness of the separator is typically ~25 μm. The gravimetric and volumetric densities are calculated based on the total weight and volume of the entire battery including current collector, electrode, and separator

charge/discharge cycles. The cell also shows remarkable stability. No obvious cell failure phenomenon was observed.

In comparison, conventional Li foil-based Li-S pouch cells can barely operate several cycles and fail upon 100 bending cycles ($r$ = 5 mm), which is ascribed to the low fatigue resistance of bare Li foil anode (Fig. 5g). This result is similar with previous published works, where most Li-S pouch cells could only be bent for one to several times. For those bent for more than 100 times, the Li-S pouch cells could typically last only for several charge/discharge cycles (Supplementary Table 4).

## Discussion

In conclusion, we have reported the first example of stable and high-energy Li-S flexible full batteries that use only 100% excess of Li, to the best of our knowledge. The cells exhibit high energy density (6.3 mW h cm$^{-2}$), high areal capacity (3 mA h cm$^{-2}$), large current density (2 mA cm$^{-2}$) and excellent cycling stability (capacity retention per cycle: 99.89% for 260 cycles) and can maintain stable charge/discharge characteristics while being repeatedly bent at small radii of curvature. The excellent electrochemical performance and mechanical flexibility of Li-S full batteries are ascribed to the unique design of fabric-typed electrodes as follows. The fabric structure of CF simultaneously endows mechanical flexibility and reduces local current density of the electrodes. More importantly, the metal coating on CF significantly stabilizes the electrode materials to reach remarkable CEs. On the anode side, Cu protective layers not only effectively prevent side reactions between Li and the CF surface, but also stabilize the SEI layer by reducing the local current density during cycling. The Cu coating renders uniform deposition of Li nanosheets instead of dendrites and leads to an average CE of 99.89% over 400 charge-discharge cycles. On the cathode side, catalytic Ni layers can effectively catalyze the reduction of soluble polysulfides, strongly capture Li$_2$S and further catalyze the decomposition back to sulfur, which enables excellent capacity retention of 99.82% over 400 cycles. As a result, the trade-off between electrochemical performance and mechanical flexibility of Li-S batteries is successfully resolved by the employment of rationally designed metallic fabrics. The achievement of mechanical and electrochemical performance sheds light on the effectiveness of the design principle. This material and electrode design principle could also be applied for other flexible and wearable energy storage devices, such as supercapacitors[53,54], Li-ion batteries[55], and Li-air batteries[56].

## Methods

**Preparation of copper-/nickel-coated carbon fabrics.** Commercially available CFs (Gas Hub Technology Pte Ltd) were immersed into a mixture of concentrated H$_2$SO$_4$/HNO$_3$ (v/v = 3:1) and then sonicated at 80 °C for 2 h. Then acid-treated CFs were rinsed using deionized (DI) water (>18 MΩ cm) several times and dried at 80 °C for 10 min. For ELD of Cu or Ni, the retreated CFs were immersed into a 4 % (v/v) [3-(methacryloyloxy) propyl] trimethoxysilane solution (solvent: 95 % EtOH, 1% acetic acid (HAc) and 4 % deionized (DI) water) for 1 h at 25 °C. After rinsing, the silanized CFs were then dipped into an aqueous mixture of [2-(methacryloyloxy) ethyl] trimethyl ammonium chloride (METAC) (20 wt%) and potassium persulfate (2 g L$^{-1}$), followed by free-radical polymerization at 80 °C for 1 h. Then, the PMETAC-coated CFs were rinsed using DI water for several times and dried at 80 °C for 10 min. After that, PMETAC-coated CFs were immersed into a 5 × 10$^{-3}$ M (NH$_4$)$_2$PdCl$_4$ aqueous solution and kept for 20 min for loading [PdCl$_4$]$^{2-}$ in a dark environment. Finally, the [PdCl$_4$]$^{2-}$ loaded CFs were individually immersed into the ELD bath of Cu or Ni for 30 min, in which a corresponding thin metal layer was deposited onto the surface of CFs. The ELD of Cu was performed in a plating bath consisting of a 1:1 (v/v) mixture of solution A and B. Solution A contains NaOH (12 g L$^{-1}$), CuSO$_4$·5H$_2$O (13 g L$^{-1}$), and KNaC$_4$H$_4$O$_6$·4H$_2$O (29 g L$^{-1}$) in DI water. Solution B is a formaldehyde (HCHO, 9.5 mL L$^{-1}$) aqueous solution. The ELD of Ni was performed in a plating bath consisting of a 10:1 (v/v) mixture of solution A and freshly prepared B. Solution A contains Ni$_2$SO$_4$·5H$_2$O (80 g L$^{-1}$), sodium citrate (40 g L$^{-1}$), and lactic acid (20 g L$^{-1}$) in DI water (pH = 7.5). Solution B is a dimethylamine borane (1.5 g L$^{-1}$) aqueous solution[44]. All ELD experiments were carried out at room temperature.

**Preparation of fabric-typed lithium anode.** The CuCF was cut into a certain size and shape to serve as the working electrode, a Celgard 2500 membrane as the separator and a lithium foil as the counter/reference electrode in the sandwiched cell. The prepared electrolyte is 1 M lithium *bis*(trifluoromethanesulfonyl)imide (LiTFSI) in a mixture solution of 1,3-dioxolane (DOL) and 1,2-dimethoxyethane (DME) (1:1, vol. ratio) with 2 wt% LiNO$_3$ additives. The cell was firstly cycled at 0–1 V (Li$^+$/Li) at 1 mA cm$^{-2}$ for five cycles to activate the cuprous oxide from cladding metallic layers. Then the required Li metal was deposited inside CuCF at 1 mA cm$^{-2}$ to obtain the Li/CuCF anode, which is easily extracted from the open cell. Here, the partial Li$^+$ was inserted into CF by passing through metallic layer and the content of inserted Li$^+$ might be easily controlled by changing the lithiation current.

**Preparation of nitrogen and sulfur heavily doped graphene.** Graphite oxide (GO) was synthesized from natural graphite powder according to the modified Hummers method[57]. Graphite oxide powder was sonicated in DI water for half an hour to yield GO solution with a concentration of 2 mg mL$^{-1}$. In brief, an aqueous dispersion of GO (2 mg mL$^{-1}$) with 2-amino thiazole in a weight ratio of 1:2 was stirred at ambient temperature to obtain amino thiazole functionalized GO. Then the mixture was sealed in a 100 mL Teflon-lined stainless-steel autoclave for hydrothermal reaction at 160 °C for 12 h. The obtained black hydrogel was immersed into DI water several times to remove the residual 2-amino thiazole and dried in the oven at 50 °C to obtain a loose NSHG sponge. For a control experiment, NG was prepared using the same procedure by adding equal amount of ethylene diamine. RGO sponge was also prepared through the same procedure without adding any dopants.

**Preparation of fabric-typed sulfur cathode.** The S$_8$@Super P nanoparticles hybrid was prepared following a melt-diffusion strategy. Commercial S$_8$ powders and Super P nanoparticles were grounded together based on the optimal weight ratio of 4:1. Then the hybrid was heated to 155 °C and maintained for 12 h. Homogeneous sulfur-containing ink was fabricated by mixing 1.05 g sulfur hybrid and 0.15 g NSHG in N-methyl-2-pyrrolidone (NMP) solvent followed by high power ultrasonication for 60 min. The as-prepared NiCF was then immersed into the concentrated ink for 10 s and removed from the ink. Finally, the NSHG/S$_8$/NiCF composite was obtained by drying in a vacuum oven at 60 °C overnight. The concentration of the ink was tuned by varying the amount of NMP to obtain different sulfur mass loading.

**Assembling of the flexible lithium-sulfur full battery cell**. The fully flexible Li-S battery cell with the encapsulation of commercial soft Al-plastic film was assembled in an argon-filled glove box using the NSHG/S$_8$/NiCF composite as cathode, a microporous polypropylene membrane as separator and the Li/CuCF composite as anode. The electrolyte of 1 M LiTFSI in DOL/DME with 2 wt% LiNO$_3$ is appropriately added according to the electrode size (20 μL cm$^{-2}$).

**Structural characterization**. The surface morphology of the electrodes was examined using field emission scanning electron microscopy (FESEM, JSM6335F, JEOL, Japan). The microstructure of the NSHG/S$_8$/NiCF composite was investigated by high resolution transmission electron microscope (HR-TEM, JEM-2011, JEOL, Japan). The NSHG/S$_8$/NiCF composite was sonicated in ethanol for 5 min and the suspension was dropped in a 200 mesh Cu grid. The elemental mapping of sulfur cathodes was performed using electron probe micro-analyzer (EPMA-1600, Shimadzu, Japan). The stress–strain curve of CuCF and NiCF was obtained using an Instron 5565 A tester. In situ resistance–strain measurements were carried out by a two-probe method through a Keithley 2400 sourcemeter. The chemical structure and composition were investigated by high-resolution X-ray photoelectron spectroscopy (XPS, Axis Ultra, Kratos) with a monochromated Al-Kα (1486.6 eV) excitation source. Raman spectroscopy was performed using a BaySpec Nomadic Raman system with a laser wavelength of 532 nm.

**Electrochemical measurements**. Stainless steel coin cells and soft-packaged cells were assembled in an Ar-filled glovebox with oxygen and moisture content < 1 ppm. In the cathodic and anodic half cells, the electrochemical performances of NSHG/S$_8$/NiCF and Li/CuCF composites were individually evaluated by galvanostatic cycling of 2032-typed coin cell with the same amount (40 μL cm$^{-2}$) of electrolyte. Galvanostatic cycling of the electrodes were conducted on Arbin and Neware battery testing systems. Cyclic voltammetry measurements were performed on a CHI660e electrochemical workstation.

**Theoretical calculation**. All theoretical calculations were performed using the Vienna ab initio simulation package code based on the first-principles of density functional theory (DFT) framework. The projector augmented wave pseudopotentials were applied to describe the electron–ion interactions. The electronic exchange correlation interaction effect was evaluated through the generalized gradient approximation with Perdew–Burke–Ernzerhof exchange–correlation function. A cut-off energy of 450 eV was employed for the plane wave basis to ensure convergence. All the structures were optimized with energy and force convergence criterions of 0.01 meV and 0.02 eV Å$^{-1}$, respectively. A vacuum slab of 20 Å was applied to exclude the interaction between all the corresponding slabs. A single layer graphene with a 4 × 4 supercell size with two-point defects was adopted as the model for non-doped reduced graphene oxide. Herein, the pyridinic nitrogen and thiophene-like sulfur were used as representative dopants for N, S heavily doped graphene. In addition, a corresponding 4 × 4 supercell size Ni slab with optimized thickness (explained in the Supplementary Table 2) was also adopted as the model for thin layer Ni-coated fabric. The vdW-DF2 correction was adopted in the simulation of absorption and decomposition processes by automatically adding the physical van der Waals interaction. The adsorption energy ($E_{ad}$) for Li$_2$S in various surface slabs are defined according to the equation: $E_{ad} = E_{total} - E_{ads} - E_{suf}$, where $E_{total}$ is the total energy of the adsorbed system, $E_{ads}$ is the energy of the adsorbate in vacuum and $E_{suf}$ is the energy of the optimized clean surface slabs. The climbing image nudged elastic band (CI-NEB) method was applied for computing the decomposition barrier ($E_{de}$) and the Li$^+$ diffusion process of Li$_2$S (Li$_2$S → LiS + Li$^+$ + $e^-$).

## Data availability

The data that supports the findings of this study are available from the corresponding author upon request.

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

## Acknowledgements

We acknowledge financial support from The Hong Kong Polytechnic University (1-YW0Z and 1-ZVK1) and RGC/NSFC (N_PolyU528/16). L.K.O. and Y.B.Q. would like to acknowledge funding from the Energy Materials and Surface Sciences Unit of the Okinawa Institute of Science and Technology Graduate University, the OIST R&D Cluster Research Program, and the OIST Proof of Concept (POC) Program.

## Author contributions

J.C. and Z.J.Z. had the original idea of the work. J.C. contributed to this work in the experimental planning, experimental measurements, data analysis, and manuscript preparation. J.S. performed the theoretical calculation. Y.M.S. and Y.C. contributed to this work in the experimental planning and manuscript preparation. L.K.O. and Y.B.Q. performed the measurement and analysis of XPS. D.W. assisted the fabrication and measurements of battery. Z.J.M., Q.Y.H., D.D.C. and G.Q.L. assisted some experimental measurements. Z.J.Z. contributed to the experimental planning, data analysis, and manuscript preparation.

## Additional information

**Competing interests:** The authors declare no competing interests.

