## [Peer Review File · Nature Communications]

Reviewers' comments:

Reviewer #1 (Remarks to the Author):

This manuscript reported the flexible and high-energy Li-S full cells with 100% oversized Li. A core approach of this work is the introduction of carbon cloth for conductive electrode substrate. The full cell performance was provided by 100% excess Li, along with the mechanical flexibility result. However, the structural characterization of the carbon cloth-based electrodes was not comprehensively conducted and the related discussion was not scientifically sound. In addition, the mechanical flexibility results were not impressive and brought some questions. Also, the electrodeposition process of Li for the fabrication of Li anodes may not be practically attractive, considering its complex/cost-consuming process. Belows are major comments of this manuscript. Considering the scientific impact of Nature Communications, this manuscript is not suitable for publication.

- In this work, the carbon fabric plays an important role as a porous substrate for the electrodes. More details on the structural characterization of carbon fabric and modified ones should be provided at each fabrication step for the electrodes, with a focus on coating uniformity over a wide range of area and cross-section direction.

- A major concern in the carbon fabric-based electrodes is the distribution of the active materials such as sulfur and lithium in the through-thickness direction. This issue becomes more significant as mass loading of electrodes is increased. For example, in Figure 5d, the sulfur loadings were increased from 1.4 to 3.2 mg cm⁻². Was the same carbon cloth used? If it is correct, the distribution of sulfur in the through-thickness direction should be provided for both cathodes. Also, what if the sulfur loading is increased to higher value? The same concern is also applied to the carbon cloth-based Li anodes. In the high mass loading electrodes, some active materials may be deposited in the form of an extra layer on top of the carbon cloth, in addition to the presence inside the carbon cloth. Under this condition, electrochemical performance and mechanical flexibilities of the carbon cloth-based electrodes may be highly different, depending on the amount of mass loading.

- In Figure S1, the morphology of the close-packed carbon fibers was observed. Under this condition, the subsequent introduction of sulfur composites through the dip coating may be challenging in terms of coating uniformity in the through-thickness direction. A cross-sectional morphology of the sulfur composite-coated cathode should be presented and the sulfur content needs to be quantified as a function of electrode thickness. Another concern is related to structural stability of the sulfur cathode upon exposure to mechanical deformation. The sulfur composite deposited on the carbon cloth fibers may be detached upon this mechanical deformation. After the bending deformation (shown in Figure 5b), the morphology of the sulfur cathode should be investigated as a function of through-thickness direction.

- The concept of combining N-doped carbons with sulfur has been reported in several publications. Although this manuscript provided a theoretical calculation for the interaction, the basic concept and effects are already known information.

- Similar to the cathodes, the Li anode also incorporated the carbon fabric substrates. The structural characterization of the Li anode should be conducted as a function of through-thickness direction. In addition, this characterization should be repeated after the bending deformation (shown in Figure 5a).

- In Figure 2F, the electrodeposited Li/CuCF showed the lower overpotential than the pristine Li foil. This electrochemical behavior should be explained in more detail.

- To verify the schematic shown in Figure 3C, additional experimental data, including the morphological results showing the deposition state at each stage, should be presented.
- According to the previous report (Adv. Mater. 2017, 29, 1605531), the electrodeposited Li tends to show inferior cell performance than the pristine Li. This issue should be addressed in the discussion of the full cell performance shown herein.
- In the preparation of flexible Li-S full cells, the authors said that a microporous membrane was used as a separator. Detailed information on the microporous membrane should be provided.
- In Figure S10, in addition to the surface morphology, the cross-sectional morphology of the electrodes after the cycling test should be provided to address the issue of thickness-directional uniformity.
- The bending results of the full cell at bending radius = 5 mm was not impressive, which may be achieved with conventional pouch Li-S cells. To highlight the mechanical flexibility of this work, a comparative study with conventional Li-S pouch cells and also the previously reported results should be conducted. In addition, more severe mechanical deformation such as the repeated folding should be conducted.

Reviewer #2 (Remarks to the Author):

This manuscript was originally submitted to Nature Energy and I was one of the reviewers. I gave a very positive and high recommendation of the Nature Energy version for publication after taking care of some minor suggestions about comparing the results with other flexible cotton derived Li-S batteries. I thought that the paper to Nature Energy had been already accepted for publication in Nature Energy. Somehow, this manuscript was transferred to Nature Communications. Again, I want to give a very positive and strong recommendation for this paper for publication in Nature Communications. The subject is very timely and of great interest to many fields. The authors designed a novel route to realizing flexible Li-S batteries. The results are solid and outstanding. The discussion part is of in depth. The reported electrochemical performance is truly outstanding in flexible Li-S batteries. The manuscript was well written. The reviewer believes that this is a paper of great quality and it is suggested that the manuscript be accepted as is.

Point-to-point Reply to reviewers' Comments

Reviewer 1:

This manuscript reported the flexible and high-energy Li-S full cells with 100% oversized Li. A core approach of this work is the introduction of carbon cloth for conductive electrode substrate. The full cell performance was provided by 100% excess Li, along with the mechanical flexibility result. However, the structural characterization of the carbon cloth-based electrodes was not comprehensively conducted, and the related discussion was not scientifically sound. In addition, the mechanical flexibility results were not impressive and brought some questions. Also, the electrodeposition process of Li for the fabrication of Li anodes may not be practically attractive, considering its complex/cost-consuming process. Bellows are major comments of this manuscript. Considering the scientific impact of Nature Communications, this manuscript is not suitable for publication.

Response: We appreciate the reviewer's comments, some of which are really helpful to improve the manuscript. According to the reviewer's suggestion, the structural characterization of metallic carbon fabric-based electrodes (anode and cathode) is systematically conducted. The related detailed discussion for electrochemical results has been explained and discussed in the main text (marked by red color). To highlight the mechanical flexibility of this work, a comparative study with conventional Li-S pouch cells using Li foils fabricated in our laboratory, and previously reported results by others have also been conducted. In addition, more severe foldable deformation is also conducted on Li-S full cells to study the mechanical stability of flexible batteries. We systematically conduct the structural/mechanical characterizations of the electrodes/cells and offer the detailed discussion of electrochemical results. With the scientific novelty in new materials design and the record-breaking device performance, we believe this manuscript is suitable for publication in Nature Communications.

But, we also believe that the reviewer might misunderstand the novelty of this work. The key of this work is NOT the introduction of carbon cloth for conductive electrode substrate. We have shown in the paper that only the carbon cloth does NOT provide stable Li-S batteries. Rather, it is the metallic carbon cloth, carbon cloth coated with metal (Cu for anode and Ni for cathode), which makes the Li-S full battery work. This is the major breakthrough of this work, as compared with previous published results using carbon cloth. The importance and novelty of metallic coated carbon cloth has been explained and discussed in detail in the introduction, discussion, and conclusion parts throughout the paper. As such, both cathode and anode show remarkable Coulombic efficiency >99.8% with very good flexibility, and the full-cell shows excellent cycling stability even with high S loading and very limited amount of Li (so very high energy density). The fundamental mechanism, materials, electrode and device have been characterized in depth. To the best of knowledge, there is no report to date on such high-energy,

stable and flexible Li-S battery, which is enabled by the metallic design of the current collector and the composite electrode structures.

Electrodeposition is widely used in various industries for preparing thin films of functional materials such as metal/alloy plating and oxide/chalcogenide semiconductors. In the industry, the mass production of Li metal is achieved by electrolyzing molten LiCl. Considering this point, the electrodeposition of Li metal is compatible with the electrolysis method in the industrial line. Currently, it is still very expensive to produce thin Li metal foils/coatings (thinner than 100 μ m) by the other process technique (evaporation, vapor deposition, extrusion or rolling), as figured out by recent literature (*Nat. Energy* 2018, 3, 267-278). It might be other simpler and cheaper methods to fabricate Li electrodes. Nevertheless, the electrodeposition of Li is never the key novelty we claimed in the paper.

1) In this work, the carbon fabric plays an important role as a porous substrate for the electrodes. More details on the structural characterization of carbon fabric and modified ones should be provided at each fabrication step for the electrodes, with a focus on coating uniformity over a wide range of area and cross-section direction.

Response: According to the reviewer's suggestion, cross-sectional SEM characterizations of both the Li/CuCF anode and the NSHG/S₈/NiCF cathode (with various method loadings) at various locations are conducted to confirm the coating uniformity over a wide range of area. At the Li/CuCF anode, Li metal prefers to deposit on the top part of CuCF due to the shorter Li⁺ diffusion pathway compared to the bottom part. With increasing areal capacity of Li/CuCF anode, the deposition of Li metal gradually extends from top to bottom due to its lithiophilic properties of CuCF. At the NSHG/S₈/NiCF cathode, the porous NiCF can uniformly adsorb the sulfur mixture over a wide range of area and cross-section direction by using vacuum infiltration method. This data is added as Supplementary Fig. 4 and Fig. 12, respectively.

Noted, this data is added as Supplementary Fig. 4.

Note that this data is added as Supplementary Fig. 12.

2) A major concern in the carbon fabric-based electrodes is the distribution of the active materials such as sulfur and lithium in the through-thickness direction. This issue becomes more significant as mass loading of electrodes is increased. For example, in Figure 5d, the sulfur loadings were increased from 1.4 to 3.2 mg cm⁻². Was the same carbon cloth used? If it is correct, the distribution of sulfur in the through-thickness direction should be provided for both cathodes. Also, what if the sulfur loading is increased to higher value? The same concern is also applied to the carbon cloth-based Li anodes. In the high mass loading electrodes, some active materials may be deposited in the form of an extra layer on top of the carbon cloth, in addition to the presence inside the carbon cloth. Under this condition, electrochemical performance and mechanical flexibilities of the carbon cloth-based electrodes may be highly different, depending on the amount of mass loading.

Response: In Figure 5d, the same metallic carbon fabric is used for both the high and low mass loading sulfur cathodes. To clearly reveal the spatial distribution of sulfur in NiCF, high-magnification SEM characterizations of NSHG/S₈/NiCF cathode is performed in the through-thickness direction as shown in the Supplementary Fig. 13 below. Combined with the top-view of SEM images (Supplementary Fig. S11c), at 1.4 mg cm⁻², the fiber surfaces are coated with sulfur. At 3.2 mg cm⁻², the space between the fibers is partially filled. As mass loading increases to 5.6 mg cm⁻², most spaces are filled up. An extra layer of sulfur may be deposited at the electrode surface if the sulfur loading is further increased. For the Li/CuCF anode, the deposition of Li metal gradually expands from top to bottom with the increase of areal capacities (Fig. 2a and Supplementary Fig. 4). An extra layer of Li metal may be deposited at the electrode surface if all the pores of CuCF are filled up.

Note that this data is added as Supplementary Fig. 13.

3) In Figure S1, the morphology of the close-packed carbon fibers was observed. Under this condition, the subsequent introduction of sulfur composites through the dip coating may be challenging in terms of coating uniformity in the through-thickness direction. A cross-sectional morphology of the sulfur composite-coated cathode should be presented, and the sulfur content needs to be quantified as a function of electrode thickness. Another concern is related to structural stability of the sulfur cathode upon exposure to mechanical deformation. The sulfur composite deposited on the carbon cloth fibers may be detached upon this mechanical deformation. After the bending deformation (shown in Figure 5b), the morphology of the sulfur cathode should be investigated as a function of through-thickness direction.

Response: Homogeneous sulfur-containing ink is fabricated by mixing 1.05 g sulfur hybrid and 0.15g NSHG in N-methyl-2-pyrrolidone (NMP) solvent followed by high power ultrasonication for 60 min. The slurry has a high viscosity range from 1k cP to 20k cP by changing the amount of solvent. The lateral size of soft NSHG is largely reduced after sonication, which can be infiltrated into the gaps between metallic fibers. Because of the 3D structure of metallized carbon fabrics, the slurry can be rapidly absorbed into the NiCF under vacuum. A cross-sectional morphology of the sulfur composite-coated cathode has been presented above, as shown in Supplementary Fig. 13. The sulfur composite is evenly absorbed into the fiber surface and the space between the fibers, indicating the uniform sulfur content distribution. To examine the detachment of sulfur composite from NiCF after bents, the morphology of the sulfur cathode after bents has also been investigated as a function of through-thickness direction. No delamination of the electrode materials from the metallic fabrics are observed (Supplementary Fig. 19).

Note that this data is added as Supplementary Fig. 19.

4) The concept of combining N-doped carbons with sulfur has been reported in several publications. Although this manuscript provided a theoretical calculation for the interaction, the basic concept and effects are already known information.

Response: The key of theoretical calculation is to show the importance of using Ni modification to the carbon fabrics. As shown in theoretical calculation, Ni surface not only offers rapid reduction/oxidation kinetics of soluble polysulfides, but also efficient absorption/decomposition of solid Li_2S . This is apparently different from and superior to previous works using only to N-doped carbons.

5) Similar to the cathodes, the Li anode also incorporated the carbon fabric substrates. The structural characterization of the Li anode should be conducted as a function of through-thickness direction. In addition, this characterization should be repeated after the bending deformation (shown in Figure 5a).

Response: As the reviewer concerned, the structural characterization of Li/CuCF anode is conducted by cross-sectional SEM technique in the through-thickness direction (Fig. 2a and Supplementary Fig. 5). Li metal is uniformly deposited into lithiophilic CuCF, reaching a Li metal layer of $\sim 80 \mu\text{m}$. For this thick sample, Li metal is plated onto the surface of metallic fibers and the space among them at the upper position of the fabric (P1, Supplementary Fig. 5). At the middle position of the fabric (P2), each metallic fiber is coated by Li metal and some gaps are filled. At the lower position (P3), no Li metal is observed.

After the bending deformation, the structural characterization of Li/CuCF anode is also performed in the Supplementary Fig. 18. No delamination of the electrode materials from the metallic fabrics are observed.

Note that this data is added as Supplementary Fig. 5.

Note that this data is added as Supplementary Fig. 18.

6) In Figure 2F, the electrodeposited Li/CuCF showed the lower overpotential than the pristine Li foil. This electrochemical behavior should be explained in more detail.

Response: It is generally accepted that mass-transfer overpotential is exacerbated at high current densities. In Figure 2F, a lithiophilic CuCF as Li hosts can provide much larger surface area than that of bare Li foil/Cu foil, as shown out in Supplementary Fig. 5. At the same applied current, CuCF with high surface area can largely reduce the local current density, resulting in a tiny mass-transfer overpotential. It is found that the Cu nanoparticles on the surface of CuCF can also confine the deposits of metallic Li and allow the continuous formation of Li nanoflakes on CuCF during the stripping/plating process. The Li nanoflakes significantly reduce the mass-transfer overpotential of Li metal by allowing much faster Li^+ transport and Li metal deposition, which lead to steadily low overpotential. Please check the second paragraph for the detailed discussion in Page 7 in the revised manuscript.

7) To verify the schematic shown in Figure 3C, additional experimental data, including the morphological results showing the deposition state at each stage, should be presented.

Response: According to the reviewer's suggestion, the morphologies results are characterized to show the deposition state at various cycling stages (I, II, and III), as shown in the Supplementary Fig. 9 below. Notably, these various cycling stages have been clearly distinguished from galvanostatic discharge profiles of Li/CuCF-LTO cell, as shown in Figure 3d. Please check the second paragraph for detailed discussion in Page 9 in the manuscript.

Note that this data is added as Supplementary Fig. 9.

8) According to the previous report (Adv. Mater. 2017, 29, 1605531), the electrodeposited Li tends to show inferior cell performance than the pristine Li. This issue should be addressed in the discussion of the full cell performance shown herein.

Response: In the work mentioned by the reviewer (Adv. Mater. 2017, 29, 1605531), the rough surface and microstructured bumps on the current collector (Cu foil) could lead to uneven Li deposits, which further amplified the surface roughness of the electrode and caused the formation

of more dendritic structures. Therefore, the electrodeposited Li metal on Cu foil showed low average Coulombic efficiency (CE: 70%) and poor electrochemical stability than pristine Li (CE: 88.3%).

On contrary to Li metal foil and electrodeposited Li/Cu foil, our Li/CuCF anode can exhibit much higher cycling stability, which can be attributed to several key factors as follows. (1) Lithiophilic CuCF with high surface areas can significantly reduce the local current density and smoothen the Li deposits. (2) The Cu coating can moderate the deposition of Li into nanosheets which give high stability. (3) Cu coating of anode prevents the interfacial side reaction between Li and the carbon fiber surface during cycling. As a result, our Li/CuCF anode exhibits a much higher average CE of 99.89 % than that of the electrodeposited Li metal on Cu foil and bare Li foil. Please check the first paragraph for the specific discussion in Page 9 in the manuscript.

9) In the preparation of flexible Li-S full cells, the authors said that a microporous membrane was used as a separator. Detailed information on the microporous membrane should be provided.

Response: Here, a microporous polypropylene film (Celgard 2500) is used to separate the two fabric electrodes for the fabrication of Li-S full cell, as we mentioned in the main text.

10) In Figure S10, in addition to the surface morphology, the cross-sectional morphology of the electrodes after the cycling test should be provided to address the issue of thickness-directional uniformity.

Response: According to the reviewer's suggestion, the cross-sectional morphology of the NSHG/S₈/NiCF cathode after cycling is provided, as shown in the Supplementary Fig. 16 below. The sulfur composite after cycling is still uniformly coated onto each fiber surface and the space between metallic fibers.

Note that this data is added as Supplementary Fig. 16.

11) The bending results of the full cell at a bending radius = 5 mm was not impressive, which may be achieved with conventional pouch Li-S cells. To highlight the mechanical flexibility of this work, a comparative study with conventional Li-S pouch cells and also the previously reported results should be conducted. In addition, more severe mechanical deformation such as the repeated folding should be conducted.

Response: To further examine the mechanical robustness of our textile-based Li-S full cells, more severe foldable deformation ($r < 1\text{mm}$) at higher current density (2.0 mA cm^{-2}) was conducted. As shown in Fig. 5f, 40 folds of the during the Li-S full batteries were carried out during the 50 charge/discharge cycles. The cell also shows remarkable stability. No obvious cell failure phenomenon was observed.

According to the reviewer's comment, a conventional Li-S pouch cell with a size of 4.0 cm^{-2} is fabricated by paring conventional Li foil with the NSHG/S₈/NiCF cathode. Upon applied 100 repeating bents at 5.0 mm radius, this conventional Li-S pouch cell only could operate three cycles and then fail, because of the low fatigue resistance of bare Li foil anode (Fig. 5g). This result is similar to previous published works, where most "so-call" flexible Li-S batteries could only be bent for one to several times. For those bent for more than 100 times, the batteries could typically last only for several charge/discharge cycles. This is summarized in Supplementary Table 4.

Note that this data is added into the revised Fig. 5.

Supplementary Table 4. Performance comparison of our Li-S full batteries and recently reported flexible Li-S batteries at the cell level.

Anode	Cathode	Li excess amount	Energy density (Wh kg ⁻¹)	Energy density (Wh L ⁻¹)	Bending cycles	Cycle life without bending	Cycle life after bent	Reference
Li/CuCF	NSHG/S ₈ /NiCF	100%	288@1.0 mA cm ⁻²	360	200@ r=5.0 mm	260	150	This work
Li foil	PVDF/CB/S ₈ /Gr@PP	4662%	94@0.45 mA cm ⁻²	48	1@ θ=90°	500	30	Adv. Mater. 27, 641-647 (2015)
Li foil	PEDOT/S ₈ /Graphene	3147%	140@3.3 mA cm ⁻²	77	1@ θ=180°	500	80	Adv. Mater. 29, 1703324 (2017)
Li foil	PVDF/MOFs/S ₈ /CNT	2757%	220@1.5 mA cm ⁻²	132	1@ θ=180°	200	60	Nat. Commun. 8, 14628 (2017)
Li foil	Graphene/S ₈ /cotton	2841%	207@1.2 mA cm ⁻²	NA	1@ θ=180°	200	50	Electrochim. Acta 246, 507-516 (2017)
Li foil	CMK-3/S ₈ /CNT	16500%	45@0.17 mA cm ⁻²	NA	1@ θ=180°	100	50	Angew. Chem. Int. Ed. Engl. 54, 10539 (2015)
Li foil	Graphene/S ₈	4230%	151@0.7 mA cm ⁻²	77.6	1@ θ=180°	100	20	J. Mater. Chem. A , 3, 9438-9445, 2015
Li foil	ABP/Ni/Gr/S ₈	2173%	266@1.78 mA cm ⁻²	NA	NA	200	NA	Electrochim. Acta 222, 1257-1266 (2016)
Li foil/CNT	PVDF/CB/S ₈ /CNT	3115%	194@2.85 mA cm ⁻²	115	100@ θ=180°	200	3	ACS Nano 9, 11342-11350 (2015)
Li foil	Li ₂ S ₆ /carbon cloth	1567%	221@1.0 mA cm ⁻²	NA	300@ r=5.0 mm	60	10	Nanomaterials 8, 90 (2018)

Reviewer 2:

This manuscript was originally submitted to Nature Energy and I was one of the reviewers. I gave a very positive and high recommendation of the Nature Energy version for publication after taking care of some minor suggestions about comparing the results with other flexible cotton derived Li-S batteries. I thought that the paper to Nature Energy had been already accepted for publication in Nature Energy. Somehow, this manuscript was transferred to Nature Communications. Again, I want to give a very positive and strong recommendation for this paper for publication in Nature Communications. The subject is very timely and of great interest to many fields. The authors designed a novel route to realizing flexible Li-S batteries. The results are solid and outstanding. The discussion part is of in depth. The reported electrochemical performance is truly outstanding in flexible Li-S batteries. The manuscript was well written. The reviewer believes that this is a paper of great quality and it is suggested that the manuscript be accepted as is.

Response: We appreciate the reviewer's time and very positive comments for our manuscript.

REVIEWERS' COMMENTS:

Reviewer #1 (Remarks to the Author):

The revised manuscript provided the well-prepared replies. However, following concerns still remain unresolved.

- The authors replied that the sulfur loading ranging from 2.4, 3.2 and 5.6 mg cm⁻² can show uniform distribution of sulfur in the cross-section of the electrodes. However, in Supplementary Fig. 12, the sulfur electrodes with the relatively low sulfur loading were shown. The author said that most spaces were filled up at the sulfur loading of 5.6 mg cm⁻². The results for the higher sulfur loading of 5.6 mg cm⁻² need to be provided. In addition, the surface morphology of the sulfur cathode should be shown to address the concern on the formation of extra layer of sulfur.
- The reviewer understand that the term of "100% oversized Li" is based on the sulfur loading of 3.2 mg cm⁻². However, in Supplementary Fig. 14, the rate capability was measured at 1.2 mg cm⁻². The rate capability at the sulfur loading of 3.2 mg cm⁻² should be provided.
- In Supplementary Table 4, additional information used for calculating the gravimetric/volumetric energy densities should be provided to ensure the validity of the values, including the data of this work.

Point-to-point Reply to reviewers' Comments

Reviewer 1:

The revised manuscript provided the well-prepared replies. However, following concerns remain unresolved.

Response: We appreciate the reviewer's comments. The followings are the details of the responses.

1) The authors replied that the sulfur loading ranging from 2.4, 3.2 and 5.6 mg cm^{-2} can show uniform distribution of sulfur in the cross-section of the electrodes. However, in Supplementary Fig. 12, the sulfur electrodes with the relatively low sulfur loading were shown. The author said that most spaces were filled up at the sulfur loading of 5.6 mg cm^{-2} . The results for the higher sulfur loading of 5.6 mg cm^{-2} need to be provided. In addition, the surface morphology of the sulfur cathode should be shown to address the concern on the formation of extra layer of sulfur.

Response: According to the reviewer's suggestion, both the enlarged SEM characterizations of the sulfur electrodes with high sulfur loadings of 5.6 mg cm^{-2} and 6.4 mg cm^{-2} are well conducted, as shown in the Supplementary Fig. 14 below. It is observed that most spaces of metallic fabrics were filled up at the sulfur loading of 5.6 mg cm^{-2} . As mass loading further increases to 6.4 mg cm^{-2} , an extra layer of sulfur is formed at the electrode surface.

Note that this data is added as Supplementary Fig. 14 in the revised Supplementary information.

2) The reviewer understands that the term of "100% oversized Li" is based on the sulfur loading of 3.2 mg cm^{-2} . However, in Supplementary Fig. 14, the rate capability was measured at 1.2 mA cm^{-2} . The rate capability at the sulfur loading of 3.2 mg cm^{-2} should be provided.

Response: According to the reviewer's suggestion, galvanostatic charge-discharge profiles and corresponding capacities at various rates of the sulfur cathodes with a sulfur loading of 3.2 mg cm^{-2} is well performed, as shown in the Supplementary Fig. 16 below.

Note that this data is added as Supplementary Fig. 16 in the revised Supplementary information.

3) In Supplementary Table 4, additional information used for calculating the gravimetric/volumetric energy densities should be provided to ensure the validity of the values, including the data of this work.

Response: According to the reviewer's suggestion, the cell weight and volume for calculating the gravimetric/volumetric energy densities of flexible lithium-sulfur full batteries are added in the Supplementary Table 4 below.

Supplementary Table 4. Performance of flexible lithium-sulfur full batteries.

Anode	Cathode	Li excess amount	Cell mass (mg cm^{-2})	Cell volume (cm^3)	Energy density (Wh kg^{-1})	Energy density (Wh L^{-1})	Bending cycles	Cycle life without bending	Cycle life after bending	Ref.
Li/CuCF	NSHG/S ₈ /NiCF	100%	21.88	0.0175	288@1.0 mA cm ⁻²	360	200@ r=5 mm	>80% @260	>90% @150	Ours
Li foil	PVDF/CB/S ₈ /Gr@PP	6813%	30.3	0.0595	99@1.1 mA cm ⁻²	50	1@ θ=90°	500	30	10

Li foil	PEDOT/S ₈ /Gr/Al	3042%	37.7	0.0575	183@3.3 mA cm ⁻²	120	1@ θ=180°	500	80	11
Li foil	PVDF/MOFs/S ₈ /CNT	2757%	33.5	0.0557	220@1.5 mAcm ⁻²	132	1@ θ=180°	200	60	12
Li foil	Gr/S ₈ /cotton	2594%	35.5	NA	227@1.2 mA cm ⁻²	NA	1@ θ=180°	200	50	13
Li foil	CMK-3/S ₈ /CNT	8542%	34.5	NA	89@0.17 mA cm ⁻²	NA	1@ θ=180°	100	50	14
Li foil	Gr/S ₈ /Al	6381%	31.8	0.0625	75@0.7 mA cm ⁻²	38.4	1@ θ=180°	100	20	15
Li foil	Gr nanotube/S ₈	8763%	28.7	NA	86@0.17 mA cm ⁻²	NA	1@ r=10 mm	500	60	16
Li foil/CNT	PVDF/S ₈ /CNT	3234%	33.9	0.0568	194@2.85 mA cm ⁻²	115	100@ θ=180°	200	3	17
Li foil	Li ₂ S ₆ /carbon cloth	1974%	46.6	NA	225@1.0 mA cm ⁻²	NA	300@ r=5 mm	100	10	18